# Of Criteria and Men—Diagnosing Atypical Parkinsonism: Towards an Algorithmic Approach

**DOI:** 10.3390/brainsci11060695

**Published:** 2021-05-25

**Authors:** Liviu Cozma, Mioara Avasilichioaei, Natalia Dima, Bogdan Ovidiu Popescu

**Affiliations:** 1Department of Clinical Neurosciences, School of Medicine, “Carol Davila” University of Medicine and Pharmacy, 020021 Bucharest, Romania; liviu.cozma@drd.umfcd.ro (L.C.); mioara-cristina.avisilichioaei@rez.umfcd.ro (M.A.); natalia-ioana.dima@rez.umfcd.ro (N.D.); 2Department of Neurology, Colentina Clinical Hospital, 20125 Bucharest, Romania; 3Laboratory of Cell Biology, Neurosciences and Experimental Myology, “Victor Babeș” National Institute of Pathology, 050096 Bucharest, Romania

**Keywords:** atypical parkinsonism, progressive supranuclear palsy, corticobasal degeneration, multiple system atrophy, dementia with Lewy bodies, PSP, CBD, MSA, DLB, diagnostic algorithm

## Abstract

Diagnosing atypical parkinsonism can be an error-exposed undertaking in the context of elaborate criteria coupled with time restraints on their comprehensive application. We conducted a retrospective, descriptive study of diagnostic accuracy among physicians at two tertiary neurology centers in Romania and developed an algorithmic tool for comparison purposes. As many as 90 patients qualified for inclusion in the study, with 77 patients actually complying with atypical parkinsonism criteria. Overall, physician-established diagnoses may be incorrect in about one-fourth of cases. The reasons for this finding span a wide range of possibilities, from terminology-related inaccuracies to criteria sophistication. A Boolean-logic algorithmic approach to diagnosis might decrease misdiagnosis rates. These findings prepare the ground for the future refinement of an algorithmic application to be fully validated in a prospective study for the benefit of patients and health professionals alike.

## 1. Introduction

Atypical parkinsonian syndromes (APS) are a group of neurodegenerative conditions that include progressive supranuclear palsy (PSP), multiple system atrophy (MSA), corticobasal degeneration (CBD), and dementia with Lewy bodies (DLB). Not only are these conditions a mix of pathologies (PSP and CBD are tauopathies, while MSA and DLB are synucleinopathies), but they also share some clinical features both within and outside of their group, with—for instance—idiopathic Parkinson’s disease [1].

Expert consensus has helped put in place formal diagnostic criteria that include core, supportive, and additional features, as well as exclusion criteria and red flags, the combinatorics of which yield different probabilistic diagnostic clusters [2,3,4,5]. Against this background, diagnosing atypical parkinsonism can be a daunting task, particularly during the early stages of the disease process [6], when the full constellation of symptoms and the relevant temporal relationships among these are not yet in place.

A study that looked specifically into the diagnostic accuracy of parkinsonian syndromes by general neurologists in Finland [7] found that atypical parkinsonism were rather underdiagnosed, while Parkinson’s disease was overdiagnosed, with as much as a quarter of diagnoses being incorrect. On the other hand, a study that looked into diagnostic accuracy by movement disorder specialists in London [8] concluded that their diagnostic yield was better than that reported for formal clinical diagnostic criteria. Where movement disorder experts are not available, or early diagnosis is essential, potential substitutes have recently been considered such as imaging-based algorithms, with or without metabolic scans, that help classify patients before a final expert diagnosis is reached [9,10].

Even though atypical parkinsonism are reportedly less frequently than idiopathic Parkinson’s disease, with a pooled prevalence of 10–18 per 100,000 inhabitants compared to an estimated total of 9.4 million people living with Parkinson’s disease globally [6,11], an accurate diagnosis is crucial from several points of view. First, an appropriately high index of suspicion might obviate the need for multiple office visits in search of a diagnosis or for unnecessary diagnostic testing. Second, treatment options would be tailored to specific atypical parkinsonism symptoms rather than to indiscriminate, syndromic dopaminergic substitution. Third, a correct diagnosis might enable patients to properly plan their lives in line with the most relevant survival prospects, join the right support group or sign up for the most appropriate clinical trial.

Data on atypical parkinsonism in Eastern Europe are scarce [12]. To the best of our knowledge, there are no data whatsoever on diagnostic accuracy or potential diagnostic algorithms. Against this background and for all the beneficial reasons listed above, it is the purpose of this article to look into the diagnostic accuracy of atypical parkinsonism in two expert centers in Romania and suggest a diagnostic algorithm that might increase the diagnostic yield in resource-strained countries. Should research results so indicate, the next steps would be the creation of a national atypical parkinsonism registry in Romania and the transposition of this fledgling algorithm into a fully functional smartphone application for the use of clinicians dealing with atypical parkinsonism. It is expected that such an application would have the necessary built-in flexibility to incorporate forthcoming revisions of the relevant criteria as well as other algorithms that have been developed for differentials with even rarer genetic entities known as ”atypical” atypical parkinsonism that phenotypically resemble PSP, CBD, MSA or DLB [13].

## 2. Materials and Methods

We conducted a retrospective, descriptive study on patients suspected of some form of atypical parkinsonism who had been seen in two tertiary neurology clinics in Bucharest, Romania (Colentina Clinical Hospital and Neuroaxis), between January 2015 and March 2021. Patients were identified in the relevant electronic databases of the two clinics using the following diagnostic codes according to the ICD-10 classification: G21.0-G23.9; G25.8-G26; G90.3-G90.9.

All patients above 18 years of age, with a sufficiently documented primary or secondary diagnosis of atypical parkinsonism—MSA, DLB, PSP, CBD or unspecified—were included in the study. Patients whose electronic files included too little information for diagnostic criteria to be applied were excluded.

Patient data were summarized in an Excel file at a level of detail such that formal diagnostic criteria for MSA, DLB, PSP and CBD could readily be applied. The relevant criteria included:“Second consensus statement on the diagnosis of multiple system atrophy” [2];“Diagnosis and management of dementia with Lewy bodies: Fourth consensus report of the DLB Consortium” [5];“Clinical Diagnosis of Progressive Supranuclear Palsy: The Movement Disorder Society Criteria” [4]; and“Criteria for the diagnosis of corticobasal degeneration” [3].

Care was taken to secure no analytic mismatch between the year of patient diagnosis and the year of the relevant criteria (old or revised) used for analysis. More specifically, PSP and DLB cases seen before the publication of revised criteria in 2017 were scrutinized against criteria valid at the time of diagnosis, namely: “Clinical research criteria for the diagnosis of progressive supranuclear palsy (Steele–Richardson–Olszweski syndrome): report on the NINDS-SPSP international workshop” and “Diagnosis and Management of dementia with Lewy bodies: third report of the DLB Consortium” [14,15].

A diagnostic algorithm was then generated in Excel enabling the most probable clinical and phenotypic diagnosis to be reached. The algorithm is a basic conditional (if–else) tool predicated on Boolean logic (true–false/1–0 values) and Boolean operators (and/or/not), the combination of which returns a diagnosis that corresponds to the aggregation of items listed in the relevant consensus criteria. Also embedded in this algorithm is the work of movement disorder specialists who developed “Multiple Allocations eXtinction” (MAX) rules enabling the choice of the best-fit PSP diagnosis from among multiple options [16]. These rules are:MAX 1 (Diagnostic Certainty): Probable > Possible > Suggestive of;MAX 2 (Temporal Order): 1st > 2nd > 3rd Diagnosis;MAX 3 (Phenotypic Hierarchy): PSP-RS > PSP-OM/PSP-PI > Other Predominance Types; andMAX 4 (MAX Hierarchy): MAX 1 > MAX 2 > MAX 3.

Given that consensus criteria consider each of the four diseases separately, for multiple allocations among these diseases we extended the use of the MAX 1 logic such that a probable diagnosis prevails over a possible one (e.g., probable PSP prevails over possible MSA). The rationale behind this choice is that a higher degree of diagnostic certainty increases specificity [17].

In summary, the algorithm was designed such that any potential user would tick off all applicable signs, symptoms and paraclinical results listed in the consensus criteria. In addition, the algorithm includes useful clues as to the correct interpretation of clinical findings. After filling in all such entries, the algorithm combines all the diagnostic elements corresponding to a particular atypical parkinsonism and returns the most probable diagnosis, including phenotype and degree of certainty. This tool is actually a mirror of the diagnostic algorithms already embedded in the criteria. Its main advantage is that it covers all four diseases at the same time and automatically returns a diagnosis without the need for the clinician to search through disparate diagnostic documents. A simplified description of the tool is available in Figure 1.

All doctor-formulated diagnoses were checked for accuracy against algorithm-generated diagnostic results. The latter were cross-checked by movement disorder specialists to remove any potential diagnostic errors. Results were processed using Excel-embedded data analysis tools. More specifically, we used descriptive statistics as applicable to continuous and categorical variables, reporting means and ranges, as well as proportions with relevant bar/pie-chart illustrations of the results.

This study was reviewed and approved by the Ethics Committees of the participant clinics.

## 3. Results

As many as 90 patients were identified who complied with all inclusion/exclusion criteria, of whom 48 were males and 42 were females. The mean age was 66.7 (±8.2 years), while the mean time to diagnosis was 2.8 years (median 2; range 0–15 years with 9 patients diagnosed within less than 1 year from onset).

The distribution of the major categories of atypical parkinsonism diagnoses identified by treating physicians, on the one hand, and by the formal algorithm, on the other hand, is illustrated in Figure 2. Overall, MSA was the most frequent diagnosis among physicians, followed quite closely by PSP, while DLB and CBD were the least frequent. MSA was also the one disease more frequently diagnosed by physicians compared to the algorithm, whereas PSP, DLB and CBD were more evenly distributed among the two diagnostic approaches. Mention should be made, at this point, of a particular diagnostic category—namely “no criteria”—signifying that physicians did not apply criteria at all to differentiate among the four diseases, while the significance of “no criteria” according to the algorithm was that no criteria were met for a diagnosis of atypical parkinsonism.

Of the 90 diagnoses formulated by treating physicians compared to algorithm-generated results, as many as 50 cases (56% of the total) were clustered as “correct”, 19 as “incomplete” and 21 as “misdiagnosed” as per Figure 3.

A more granular analysis of these clusters revealed that diagnosis among physicians was not so much a “yes” or “no” undertaking, but rather a matter of degrees of precision. As such, within the three major categories of accuracy identified above, one could note several subentries, as per Table 1.

An analysis of the number of cases pertaining to each subentry showed that, within the “correct” cluster, only 8 cases (9% of the total) were fully characterized as per formal criteria requirements, while most other diagnoses failed to include all the relevant phenotype/certainty-related details. At the other end of the spectrum, as many as 13 cases (14% of the total) were misidentified as atypical parkinsonian syndromes. A full description of the distribution of cases per degree of diagnostic precision is rendered in Figure 4.

Looking into diagnostic precision per type of atypical parkinsonism, one could note that PSP covered the full spectrum of degrees of precision, whereas DLB spanned a more limited range. MSA and CBD both fell in between these two possibilities and were closer to DLB than to PSP, as illustrated in Figure 5, Figure 6, Figure 7 and Figure 8.

With specific reference to each of the four diseases, one could note that, in the case of MSA, for instance, diagnosis was most often correct, with missing details on the certainty level being the most common diagnostic imperfection (Figure 5).

In the case of DLB, the proportion of incomplete and incorrect diagnoses was higher than the proportion of correct diagnoses. As with MSA, failing to indicate the level of certainty associated with the diagnosis was the most frequent type of diagnostic imprecision (Figure 6).

Interestingly, while most widely dispersed in terms of degree of precision, most PSP diagnoses were in the middle of the diagnostic precision spectrum as either incomplete or correct but failing to indicate both the phenotype and the level of certainty (Figure 7).

Last but not least, CBD had a narrower dispersion in terms of degree of diagnostic precision and a higher proportion of incomplete/overlap diagnoses. Also, CBD was the one disease category with the fewest cases of misdiagnoses when no criteria for any atypical parkinsonism were met (Figure 8).

Data also indicated that diagnostic accuracy was fairly nuanced per physician category (junior and senior) as per Table 2 below.

## 4. Discussion

Diagnosing atypical parkinsonism ante-mortem is—for the most part—a clinical enterprise based mostly on history and the neurologic exam. Diagnostic conundrums are a frequent occurrence given that signs and symptoms overlap not only with other kinds of parkinsonism (i.e., idiopathic; genetic; secondary), but also within the family of atypical parkinsonian syndromes. Establishing a firm—though not yet definitive pathological—diagnosis may take years, imposing a high burden on both patients and health systems [18], particularly in resource-limited countries. Against this background, we deemed it important to evaluate diagnostic pitfalls and develop diagnostic tools for the use of general neurologists.

To the best of our knowledge, the study herein presented is the first original research into atypical parkinsonism diagnostic accuracy in Romania. In addition, to the best of our knowledge, no diagnostic algorithm has hitherto been put together that could help improve the diagnosis of atypical parkinsonism. The results of this study indicate that diagnostic work can be improved at several levels as will be detailed below.

First, while a Boolean-logic tool might be in place, its use is predicated on the full and correct understanding of the particular meaning of signs and symptoms against which a diagnosis is checked. Our initial data-combing through the clinic’s database at the time of patient inclusion revealed that clinicians were not always on the lookout for all signs and symptoms associated with atypical parkinsonism. Nor was one and the same term in the criteria described consistently or comprehensively across clinicians. To give only a few examples, we identified some error-prone areas as follows:‑vertical gaze palsy was either over- or under-interpreted (saccade initiation instead of velocity being considered as a marker of the condition; isolated and barely perceptible limitation of up-gaze being over-interpreted; vertical gaze palsy being considered a PSP-pathognomonic sign without due consideration of “atypical” atypical parkinsonism or other diseases);‑macro square-wave jerks were never identified/mentioned;‑myoclonus and dystonia were under-represented;‑orthostatic hypotension was neither properly defined at 3 min nor fully quantified against systolic and diastolic specific criteria; consideration of other potential and more frequent causes of hypotension (e.g., diabetes, medication, hypovolemic states) was not fully documented;‑alternative/exclusionary causes of urinary dysautonomia (prostatic disease, diabetes, pelvic-floor muscle weakness in elderly women) were not always identified/described;‑inquiries were seldom made about associated erectile dysfunction;‑cognitive deficits were not fully qualified (subdomain-specific limitations were not identified);‑fluctuations in cognition and hallucinations were not fully described against criteria requirements; and‑levodopa trials did not comply with the full criteria specifications for dosage and responsiveness.

Second, the analysis of our data indicated a different disease frequency compared to data reported in the literature. More specifically, physicians in the study identified MSA most frequently, with PSP as the second most frequent diagnosis, followed at a considerable distance by DLB and CBD. In contrast, wider prevalence reports point to DLB as a fairly common disease in the elderly (400 cases per 100,000 persons), with MSA and PSP both having a prevalence of 5 to 10 per 100,000 persons, while the prevalence of CBD is about 1 per 100,000 [19]. The reason for this difference may be related to both the size of the lot under consideration (90 patients) and zonal limitations (one city). Thus, while the two tertiary clinics involved in the study are fairly representative for expertise in movement disorders, their location in the capital city of Bucharest might make it difficult for potential patients across the country to call upon their expert services. One other potential reason is that patients with more prominent dementia and little parkinsonism might first access psychiatry services instead of neurology clinics. Moreover, dementia is most often associated with Alzheimer’s disease with little follow-up in search of other potential etiologies. Last but not least, dementia-related codes as per the electronic record system were not investigated, which may have been a limitation of our methodology, although the combination with parkinsonism or other manifestations as secondary diagnoses should have helped with the identification of less obvious cases. A similar comment could be made with respect to idiopathic Parkinson’s disease, which was not included in the original diagnostic code search. Had this been the case, disease frequencies might have looked different given that atypical parkinsonism is often mistaken for idiopathic Parkinson’s disease [7].

Third, this physician-algorithm comparative study revealed a fairly high rate of correct diagnoses at 56% of the total. However, as many as 23% of all cases were misdiagnosed, which is a rather significant proportion of failure. The remainder (21%) is made up of incomplete diagnoses reflecting perhaps the grey zones of clinical overlap. Diagnostic dilemmas are frequent both among senior doctors and among junior specialists. The reasons for this overall picture are, perhaps, manifold, as follows:consensus criteria are rather lengthy and difficult to follow; they also are not readily available in one place/on-the-go;time/resource limitations during one or several office visits may detract from full consideration of the criteria;symptom-/sign-related terms are not always fully defined and usage is rather specialist-oriented;criteria include several areas of overlap among the four diseases with diagnostic choices difficult to make when symptoms and signs are barely manifest;some consensus criteria (MSA and DLB) do not include absolute exclusion criteria that might have helped rule out potential alternatives at an earlier stage; in fact, the MSA criteria have been fairly recently criticized [20] in light of the significant mismatch between clinical and pathological diagnoses, with a call for criteria revision that is currently under way; as far as DLB is concerned, the debate is not yet over as to a real biological difference between DLB and Parkinson’s disease dementia as opposed to both entities being situated on a disease/pathologic continuum [21]; andphysicians do not seem to be applying the mandatory or context-dependent exclusion criteria in place for PSP and CBD.

Fourth, it could be noted from the analysis that there were many nuances within the major diagnostic categories. Among these, the most frequent within the “correct” category seemed to be a failure to fully characterize the disease by including specifications on phenotypes and degrees of certainty. One of the reasons for this failure might be that physicians tend to ignore treatment-irrelevant details and do not think in broader research-related terms as opposed to a more limited, immediate, clinical relevance. However, as has amply been made clear, phenotype and degree of certainty differentiation are critical not only in terms of prognosis but also in terms of clinical trial recruitment [4,22].

To add finesse to the point highlighted above, one might also note that PSP was the one disease with the widest dispersion of diagnostic precision and with most instances of failure to qualify the relevant phenotype. This may appear paradoxical, given that PSP consensus criteria have been fairly recently revised specifically with a view to enhancing diagnostic precision. In our local practice, however, more specific criteria did not translate into better phenotypic identification. One potential reason for this unexpected outcome might be that real-life clinical practice does not allow practitioners to go at length through all the relevant items.

Along the spectrum of tauopathies, CBD criteria are less sophisticated than PSP criteria; however, most CBD diagnoses fell within the “incomplete” category deriving from instances of PSP/CBD overlaps. The reason for this might be that no criteria are available to delineate between these two phenotypically and pathologically similar diseases. In fact, the degree of overlap can be such that some specialists in the field question the clinical utility of a separation between the two [23]. Furthermore, this has led to the proposal that a joint “tauopathy 4R” diagnosis be used in cases of significant overlap [4,24]. CBD is also the least frequent correct diagnosis and misdiagnosis when no criteria for CBD are met. This could be explained by physicians rarely taking CBD into account, except only when a striking CBS phenotype is at stake, coupled with a failure to apply criteria for other CBD phenotypes.

Compared to the wide PSP diagnostic dispersion, MSA seemed to be most often correctly diagnosed. The reason for this might be a combination of criteria simplicity and high frequency of dysautonomia in the elderly, which raises a red flag for the early consideration of an MSA diagnosis. The latter might also be the reason why other atypical parkinsonism cases are often categorized as MSA. Along the alpha-synucleinopathy spectrum, DLB was associated with the highest rate of incomplete and incorrect diagnoses (although this tentative analysis might be skewed by the limited number of DLB cases under consideration—8 cases in total). Still, one explanation might derive from a failure to identify cognitive fluctuations and construe hallucinations as defined in the relevant criteria.

For all these reasons, we have thought it important to make a first attempt at developing an algorithmic diagnostic tool for atypical parkinsonism that could strike the right balance between scientific complexity and simplicity of use for the benefit of regular clinicians. The encouraging results of this retrospective, file-based, study lay the groundwork for future efforts at turning this currently Excel-based tool into a full-fledged smart application that will need proper validation in a properly sampled, multicenter prospective study. It is expected that the code of such an application will be flexible enough to accommodate any forthcoming criteria revisions as well as other relevant criteria for differentials with related groups of diseases. Moreover, we trust the results of this study could help configure a national registry of atypical parkinsonism in the interests of patients, clinicians and researchers alike.

The simple diagnostic approach considered in this article is set against a research background that is rife with attempts at artificial intelligence/machine learning-based diagnostic tools. Indeed, efforts are being made for automated detection of eye movement or gait patterns [25,26]—to give only two examples. For the time being, however, such efforts address the automated recognition of disparate signs and symptoms (saccades, gait, etc.) included in the consensus criteria. Such machine outputs are yet to be perfected and still rely on human trial-and-error input for satisfactory sign/symptom recognition. The merit of our physician-driven diagnostic approach is that it considers all atypical parkinsonism consensus criteria simultaneously, allowing for the best clinical choice among—sometimes overlapping—diagnostic probabilities. These pathologies are too intricate and data-laden for any single machine-based algorithm to capture their underlying complexity and return an accurate, clinical, ante-mortem diagnosis.

While it could be argued that machines might replace physicians when it comes to diagnostic tools in the future, the debate is still on as to whether the replacement will be total, partial or rather symbiotic [27,28].

At the same time, the approach herein aggregated has its limitations, and there are multiple levels where it may be imprecise in regard to diagnostic accuracy.

First, it is important to recognize that the application of these criteria might yield divergent results depending on the stage of the illness. For example, a validation study found that applying the criteria for probable PSP in the first 3 years of disease resulted in a sensitivity of 33.3% and a specificity of 81.6%, while applying those same criteria more than 3 years after onset had a sensitivity of 81.1% and a specificity of 91.4% [17]. Since our approach tries to apply these criteria as best as possible, this primarily means that they also mirror the imprecision inherent in the dynamics of the disease before the relevant clinical picture is fully blown. In particular, the patients in our study had a mean time to diagnosis of 2.8 years, with a median of 2 years—comparable to reports in other studies on the topic [29].

Second, when combing the four sets of criteria for MSA, DLB, PSP and CBD, yet another level of diagnostic imprecision comes up, inherent in the many areas of overlap among particular criteria subsets, coupled with a step-wise consideration of diagnostic certainty. Our “final” diagnosis was the one indicated by the highest degree of certainty (“probable” > “possible” > “suggestive of”). This is because a probable diagnosis increases specificity for that particular disease (e.g., more than 3 years into the disease, “probable” PSP has a specificity of 91.4%, while “suggestive of” PSP has a specificity of 31.4%) [17].

Third, the entire approach is predicated on how criteria are understood and applied by their very users. However, a tool that aggregates all criteria in one place might direct physician efforts to the comprehensive interpretation and consideration of those criteria as compared to their application based solely on human memory. As such, there would be less room for diagnostic error, at least with respect to the application of formal consensus criteria.

Finally, it should also be borne in mind that a “definitive” diagnosis can only be made based on brain biopsies and that, for the time being, the intricacies of neurodegeneration only allow for, at most, a “probable” diagnosis to be made during regular clinical practice.

## 5. Conclusions

This first-of-its-kind study looked into diagnostic difficulties associated with atypical parkinsonism and found that misdiagnosis was rather frequent in about one-fourth of cases; on the other hand, diagnosis was also found to be correct—albeit not in full—in about half of all cases. These findings are in line with the hypothesis that atypical parkinsonism is difficult to diagnose correctly for a variety of reasons; we have therefore attempted to correct diagnostic imperfections through an algorithmic approach. At a time when technology and medicine are more symbiotic than ever before, we will continue to improve upon the results of this work by developing an easy-to-use diagnostic application and an associated national registry for the purposes of better diagnosis, better resource allocation and more participation in clinical trials aimed at disease-modifying therapies.

## Figures and Tables

**Figure 1 brainsci-11-00695-f001:**
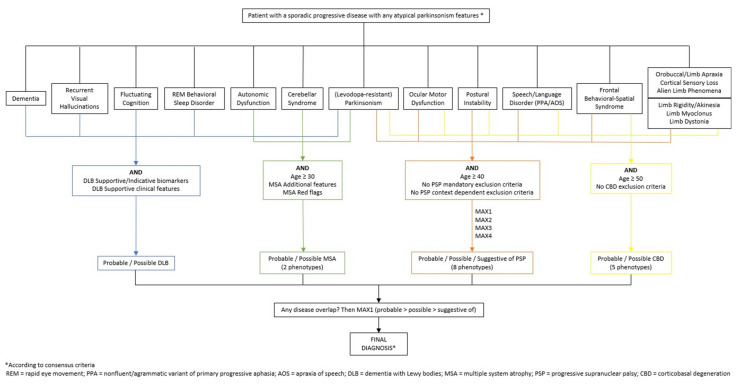
Simplified Decision Tree.

**Figure 2 brainsci-11-00695-f002:**
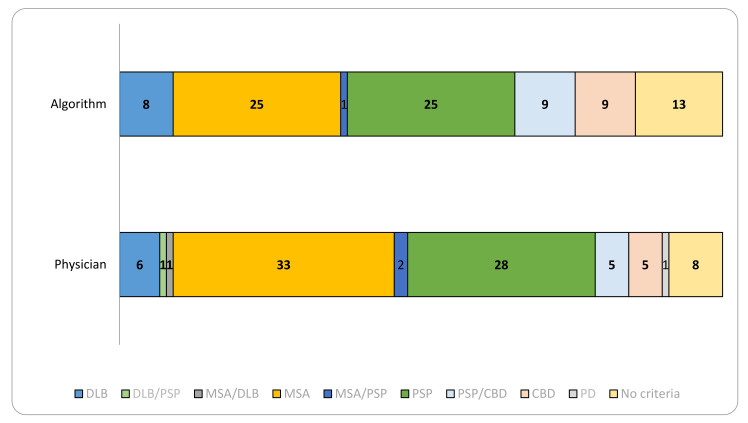
Algorithm versus physician diagnosis of atypical parkinsonian syndromes.

**Figure 3 brainsci-11-00695-f003:**
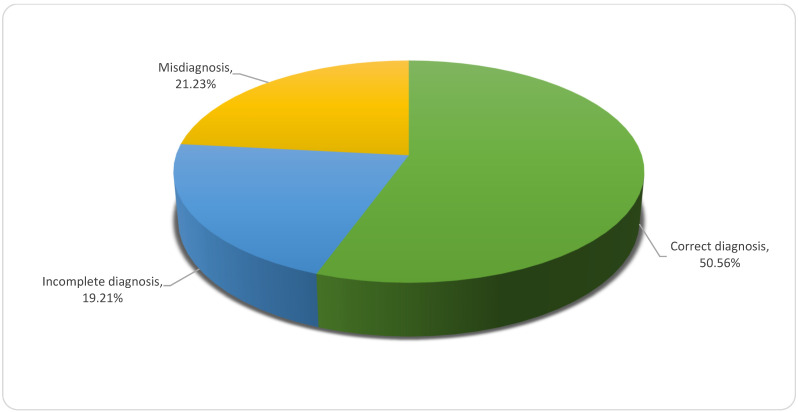
Overall diagnostic accuracy of atypical parkinsonian syndromes.

**Figure 4 brainsci-11-00695-f004:**
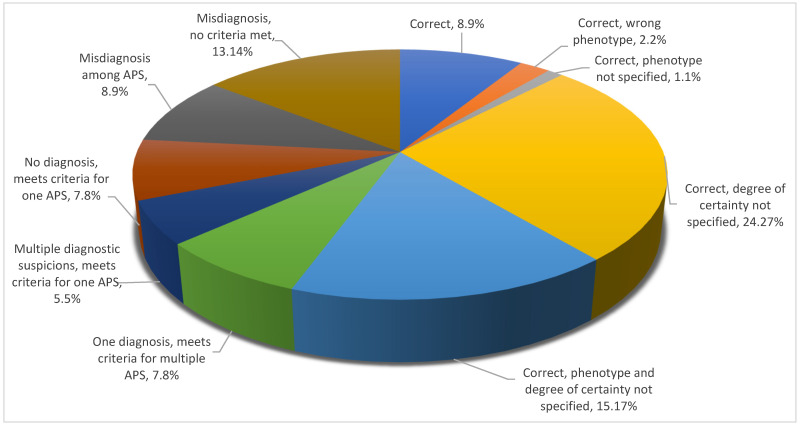
Distribution of diagnostic accuracy per degrees of precision.

**Figure 5 brainsci-11-00695-f005:**
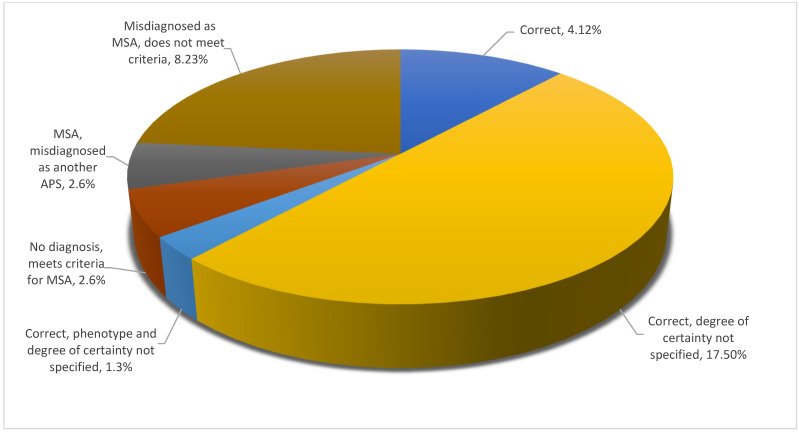
Diagnostic precision per type of atypical parkinsonism: MSA.

**Figure 6 brainsci-11-00695-f006:**
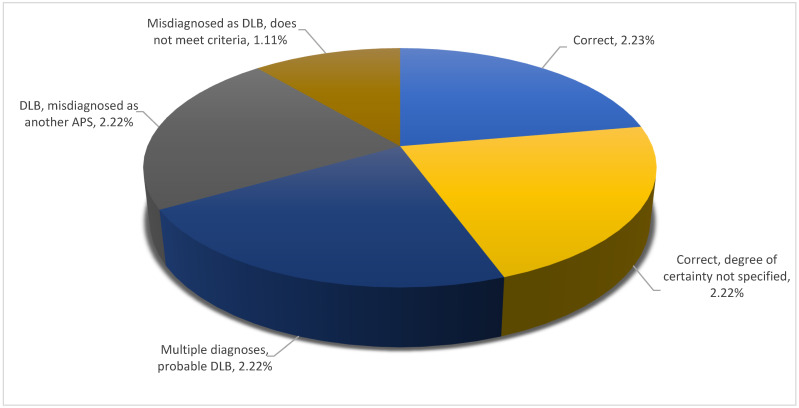
Diagnostic precision per type of atypical parkinsonism: DLB.

**Figure 7 brainsci-11-00695-f007:**
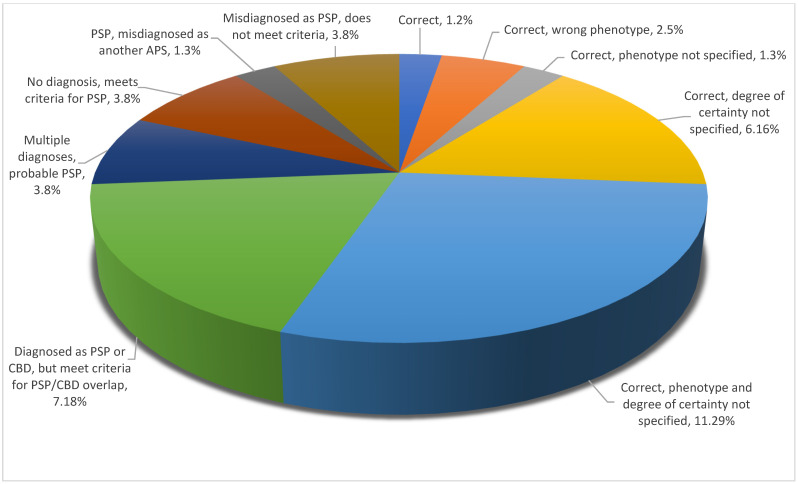
Diagnostic precision per type of atypical parkinsonism: PSP.

**Figure 8 brainsci-11-00695-f008:**
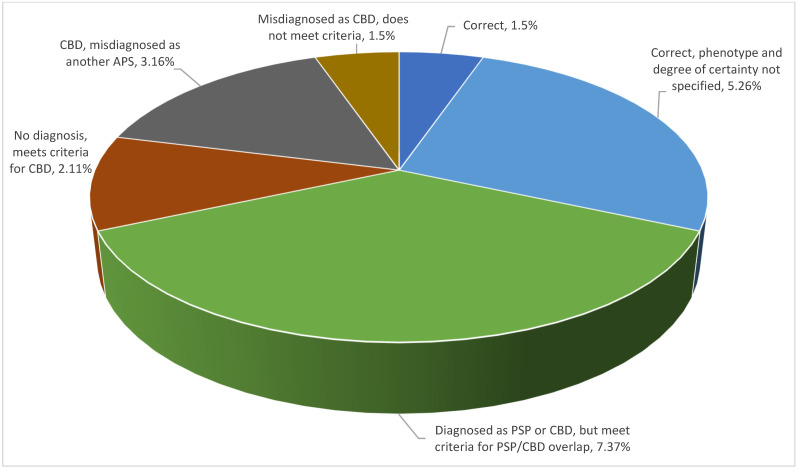
Diagnostic precision per type of atypical parkinsonism: CBD.

**Table 1 brainsci-11-00695-t001:** Diagnostic degrees of precision.

I—CORRECT DIAGNOSIS
1- fully correct (disease, phenotype and degree of certainty all included)
2- correct disease identified but wrong phenotype indicated
3- correct disease and degree of certainty indicated but no phenotype identified
4- correct disease and phenotype identified but no degree of certainty indicated
5- correct disease but neither phenotype nor degree of certainty indicated
**II—INCOMPLETE DIAGNOSIS**
6- one disease identified but formal criteria algorithm indicated disease overlap
7- multiple diseases identified but formal criteria algorithm indicated one disease
8- indeterminate atypical parkinsonian syndrome identified but formal criteria algorithm indicated a particular disease
**III—MISDIAGNOSIS**
9- incorrect disease; formal criteria algorithm indicates another atypical parkinsonism
10- incorrect disease; formal criteria algorithm indicates no atypical parkinsonism

**Table 2 brainsci-11-00695-t002:** Diagnostic accuracy among senior and junior physicians.

	Correct	Incomplete	Misdiagnosis	Total
**Seniors**	30 (55%)	14 (25%)	11 (20%)	55
**Juniors**	20 (57%)	5 (14%)	10 (29%)	35

## Data Availability

The data presented in this study are available on request from the corresponding author. The data are not publicly available due to privacy issues.

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
