# Peer review of "Of Criteria and Men—Diagnosing Atypical Parkinsonism: Towards an Algorithmic Approach"

_brainsci, 2021, doi:10.3390/brainsci11060695_

Round 1

Reviewer 1 Report

The present study entitled " Of Criteria and Men – Diagnosing Atypical Parkinsonism: Towards an Algorithmic Approach" by Cozma et al. describes retrospective evaluation of the reliability of diagnostic approaches and criteria for Atypical Parkinsonism used by the physicians, by comparing PSP, MSA, DLB and CBD cases. This study mainly emphasizes on scrutinizing the misdiagnosis or errors at physicians end. There are some minor comments/queries listed below:

Line#52. Statement by authors "doctors-shopping in search of a diagnosis" needs to be corrected to more meaningful sense.

In Materials and Methods section, authors need to mention the diagnostic scales and/or international standard diagnostic criteria used by physicians for PSP, MSA, DLB and CBD.

Sample size of 90 is not satisfactory, therefore authors need to increase the number of patients in this study.

Discussion section lacks proper citations, so authors need to re-write it with appropriate citations.

If possible, authors should provide additional description about the statistical methods used in their study.

Author Response

We thank the reviewer for his kind contribution. Please find our replies below.

  1. Line#52. Statement by authors "doctors-shopping in search of a diagnosis" needs to be corrected to more meaningful sense.

As suggested, we replaced “doctor-shopping” with “multiple office visits” (line 52).

  1. In Materials and Methods section, authors need to mention the diagnostic scales and/or international standard diagnostic criteria used by physicians for PSP, MSA, DLB and CBD.

We have modified the Materials and Methods section (lines 82-97) to take account of the reviewer’s comment as follows:

“Patient data were summarized in an Excel file at a level of detail such that formal diagnostic criteria for MSA, DLB, PSP and CBD could readily be applied. The relevant criteria included:

  • “Second consensus statement on the diagnosis of multiple system atrophy” [2]
  • “Diagnosis and management of dementia with Lewy bodies: Fourth consensus report of the DLB Consortium” [5]
  • “Clinical Diagnosis of Progressive Supranuclear Palsy: The Movement Disorder Society Criteria” [4]
  • “Criteria for the diagnosis of corticobasal degeneration” [3]

Care was taken to secure no analytic mismatch between the year of patient diagnosis and the year of the relevant criteria (old or revised) used for analysis. More specifically, PSP and DLB cases seen before the publication of revised criteria in 2017, were scrutinized against criteria valid at the time of diagnosis, namely: “Clinical research criteria for the diagnosis of progressive supranuclear palsy (Steele-Richardson-Olszweski syndrome): report on the NINDS-SPSP international workshop” and “Diagnosis and Management of dementia with Lewy bodies: third report of the DLB Consortium” [14, 15].”

We have also included the following paragraph (lines 123-131) to better summarize the workings of the algorithm:

“In summary, the algorithm was designed such that any potential user would tick off all applicable signs, symptoms and paraclinical results listed in the consensus criteria. Also, the algorithm includes useful clues as to the correct interpretation of clinical findings. After filling in all such entries, the algorithm combines all the diagnostic elements corresponding to a particular atypical parkinsonism and returns the most probable diagnosis, including phenotype and degree of certainty. This tool is actually a mirror of the diagnostic algorithms already embedded in the criteria. Its main advantage is that it covers all four diseases at the same time and automatically returns a diagnosis without the need for the clinician to search through disparate diagnostic documents.”

As regards diagnostic scales, please note that no such scales are in place specifically for the diagnosis of atypical parkinsonisms. Evaluation scales, on the other hand, are indeed available, but these were not within the scope of this article, which was strictly concerned with the application of consensus criteria.  

  1. Sample size of 90 is not satisfactory, therefore authors need to increase the number of patients in this study.

We agree with the reviewer that the analysis could benefit from a larger number of patients, including a proper sample once a national registry is in place. However, as mentioned in our article, no such national registries are available in Romania; furthermore, it is one of the purposes of our study to serve as a stepping stone to the creation of one such registry.

For the moment we have used the data available in the two referral clinics that had enough information to permit a satisfactory application of the relevant criteria for all the four atypical parkinsonisms considered. Also, given the rather low prevalence of these diseases in the general population, it would be very difficult to increase the number of patients.

More specifically, with a reported prevalence of about 5 per 100.000 persons for MSA and PSP, respectively, and 1 per 100.000 persons for CBD (DOI: 10.3238/arztebl.2016.0061) the prevalence of these 3 atypical parkinsonisms in the whole population of Romania would, in theory, amount to a prevalence of about 1980 persons. Given these numbers and considering that many patients go undiagnosed, we would not consider the number of patients included in our study to be negligible, although of course we would have liked for it to be as high as possible.

Even very large studies, such as that of Jabbari et al (doi:10.1001/jamaneurol.2019.4347) covering a natural history cohort, recruited a number of only 222 patients with atypical parkinsonism from 7 sites in the United Kingdom over 3.3 years.

Also, Buril et al. (https://doi.org/10.1371/journal.pone.0246342) reported a number of only 1218 visits from patients with a diagnosis of atypical parkinsonism in the entire Czech Republic. Other epidemiological studies also report very low number of cases (DOI: 10.1212/wnl.49.5.1284 / https://doi.org/10.3389/fneur.2020.00180 / DOI: 10.1002/mds.22966).

  1. Discussion section lacks proper citations, so authors need to re-write it with appropriate citations.

The discussion section gives an overview of the reasons that might explain the results of the study with appropriate reference to the relevant literature where available. Similar studies looking into diagnostic accuracy did not provide details on error-prone areas in applying the  diagnostic criteria. Relevant references are cited in our article. Moreover, we did not identify any studies that particularly looked at how the criteria for atypical parkinsonisms are applied in clinical practice. As such, we would be grateful for more specific details on how to improve this section.

  1. If possible, authors should provide additional description about the statistical methods used in their study.

As per recommendation, we added the following phrase: “More specifically, we used descriptive statistics as applicable to continuous and categorical variables, reporting means and ranges, as well as proportions with relevant bar/pie-chart illustrations of the results.” (lines 135-137).

Please note that we did not opt for running a McNemar’s exact test (which would have befitted a comparison of categorical outputs between two groups with sparse data) given that one of the terms of the comparison (i.e. the algorithm) did not have a term of comparison outside of itself (i.e. the algorithm we defined returned 100% correct results having been designed to match the diagnostic criteria as defined in the consensus papers). Under the circumstances, with 100% correct results and no errors as per algorithm results, statistical significance would have followed by default.

Reviewer 2 Report

The authors propose a simple algorithmic approach for the diagnosis of Parkinson’s disease. The paper is poorly prepared from the methodological and technical point of view. Modern approaches for Parkinson disease diagnostics use artificial intelligence and machine learning methods (see, for example, doi:10.1016/j.patrec.2019.04.005, oi:10.1109/ACCESS.2020.2995737, doi:10.1109/JBHI.2019.2891729). Unfortunately, such simplistic approaches as the one proposed in this paper do not work. The presented algorithm is not described in a technical way (using pseudocode or using a decision tree). The algorithm does not consider the stages of Parkinson's disease, which is a major flaw. The achieved accuracy is low. The results are not compared with the results of other works. The conclusions are not convincing.

Author Response

We thank the reviewer for her/his straightforward comments. We acknowledge our study has several limitations, some of which we tried to address in the article.

We would like to mention that Parkinson’s disease is not the subject matter of our study. Instead, we focused specifically on diagnosing atypical parkinsonisms (e.g. multiple system atrophy, dementia with Lewy bodies, progressive supranuclear palsy and corticobasal degeneration). We recognize that Parkinson’s disease should be considered in the differential diagnosis of atypical parkinsonisms, but correctly applying the diagnostic criteria for the latter should automatically exclude a diagnosis of Parkinson’s disease in most instances, even taking into consideration all the stages of the disease. For example, many inclusion criteria for atypical parkinsonisms are absolute exclusion criteria for Parkinson’s disease.  

While the reviewer is very much right that modern approaches to the diagnosis of various diseases use artificial intelligence and machine learning techniques, the use of such methods is limited in the diagnosis of atypical parkinsonisms and is currently very far from being relevant in routine clinical practice, as well as in the implementation of future diagnostic criteria (i.e. multiple system atrophy criteria currently under revision do not propose the use of AI/machine learning). While this kind of technological advances might and will probably prove to be very useful in the future, we believe there is a wide gap between research and clinical practice where simpler algorithms might be of much practical and immediate use. As shown by our study, the problem with the diagnosis of atypical parkinsonism lies in the misapplication of diagnostic criteria. This is not surprising given their complexity, both in isolation and, even more so, in combination. Also, given the very low prevalence of these diseases and their diagnostic difficulty compared to Parkinson’s disease, such advanced techniques might not make their way into regular diagnosis any time soon.

The reviewer is right in pointing out that this algorithm is simple. It was our intention to propose a simple algorithm for the use of busy regular clinicians, where most diagnostic errors are encountered. Please also take into consideration that our analysis was conducted in tertiary referral centers, where clinicians are actually more frequently exposed to such rare diseases.

In response to the reviewer’s indications, we have tried to improve upon our materials and methods section and included an extra paragraph explaining the content and use of the algorithm (lines 123-131).

We tried to compare our results to those of other studies, but could not identify works similar in scope and intent to ours. More specifically, we did not identify any studies that particularly looked at how the criteria for atypical parkinsonisms are applied in clinical practice. Studies looking into diagnostic accuracy did not provide details on error-prone areas in applying the relevant diagnostic criteria. Where applicable, relevant references are cited in our article.

We did not clearly understand what the reviewer meant by low achieved accuracy. We would be grateful for more details so as to be able to improve our article. 

Round 2

Reviewer 1 Report

Thanks for re-submitting the manuscript with relevant corrections.

Author Response

We do theank the reviewer for the very useful help to improve the manuscript.

Reviewer 2 Report

The authors have provided a rather lengthy response letter, however, only minor modifications of the paper itself were done, which did not solve my concerns. I must conclude that, unfortunately, the authors are not willing to do a major revision of this manuscript.

Author Response

We do thank the reviewer for the very helpful comments. We did majorly revised the manuscript and now we hope all the issues raised are properly addressed.